# CONFIDENCE IS NOT COMPETENCE

## ABSTRACT

Large language models (LLMs) often exhibit a puzzling disconnect between their asserted confidence and actual problem-solving competence. We offer a mechanistic account of this decoupling by analyzing the geometry of internal states across two phases: pre-generative assessment and solution execution. A simple linear probe decodes the internal "solvability belief" of a model, revealing a well-ordered belief axis that generalizes across model families and across math, code, planning, and logic tasks. Yet, the geometries diverge; although belief is linearly decodable, the assessment manifold has high linear effective dimensionality as measured from the principal components, while the subsequent reasoning trace evolves on a much lower-dimensional manifold. This sharp reduction in geometric complexity from thought to action mechanistically explains the confidence–competence gap. Causal interventions that steer representations along the belief axis leave final solutions unchanged, indicating that linear nudges in the complex assessment space do not control the constrained dynamics of execution. We thus uncover a two-system architecture: a geometrically complex assessor feeding a geometrically simple executor. These results challenge the assumption that decodable beliefs are actionable levers, instead arguing for interventions that target the procedural dynamics of execution rather than the high-level geometry of assessment.

## 1 INTRODUCTION

A major challenge in using LLMs Team et al. (2025); Grattafiori et al. (2024); Jiang et al. (2023); Qwen et al. (2025) in critical areas is that they can produce answers that sound fluent and convincing but are actually wrong, while showing high confidence Maharana et al. (2025); Xiong et al. (2023); Ren et al. (2023). This disconnect between apparent confidence and actual competence is not merely an academic curiosity; it is a fundamental reliability problem that undermines trust in scientific discovery, medical diagnostics, and logical reasoning systems powered by these models. While prior work has documented this **confidence-competence gap** from a behavioral perspective Singh et al. (2023) and the general unreliability of interventional methods Tan et al. (2025); Braun et al. (2025), a core question remains unanswered: why does a model's internal cognitive state appear decoupled from its final actions? This paper moves beyond behavioral observation to provide the first mechanistic account of this phenomenon.

A guiding hypothesis in mechanistic interpretability is that if we can isolate the neural representation of a model's belief about whether a problem is solvable, we can understand and perhaps directly control its competence Marks & Tegmark (2024); Dunefsky & Cohan (2025). Following this intuitive path, however, leads to a deep paradox. Uncovering such representations requires exquisite experimental control to isolate the faint signal of belief from the high-dimensional noise of heuristics. In contrast to prior work, we employ a meticulously designed experimental framework, utilizing length-controlled datasets of non-trivial reasoning problems across multiple model families (Qwen-2.5 Qwen et al. (2025), Gemma-3 Team et al. (2025), Mistral Jiang et al. (2023)) and probing techniques, to ensure we are capturing a true representation of belief, not a superficial correlate. Yet, this rigorous isolation effort did not resolve the problem; it sharpened the paradox, revealing a functional decoupling between assessment and execution.

To dissect this paradox, our investigation proceeds in three logical stages. First, we establish the existence and nature of a latent **Belief state**, an internal, pre-generative assessment of task solvability, distinct from token-level confidence metrics like perplexity or softmax scores. Using a suite of linear

probes and representational similarity measures (CKA) Kornblith et al. (2019); Zhou et al. (2024); Murphy et al. (2024), we demonstrate that a model's belief in its own problem-solving ability is robustly and linearly encoded across a diverse set of math, code, and logic tasks. Second, we subject this validated belief state to a direct causal test. We intervene on this representation, using targeted steering vectors to forcefully alter the model's belief from *unsolvable* to *solvable* and vice-versa Li et al. (2023); Rimsky et al. (2023); Lee et al. (2024); Turner et al. (2023). We test the central hypothesis: is belief an active participant in reasoning, or a passive observer? Finally, having observed a low casual influence, we analyze the geometry of the representations. Using principal component analysis, we show a geometric difference Park et al. (2023); Marks & Tegmark (2023); Balestriero et al. (2023); Konen et al. (2024) between the high-dimensional manifold of **Assessment** (belief) and the narrow, low-dimensional manifold of **Execution** (competence). Lastly, we perform a variance analysis to identify the cognitive collapse at the **transition point** between the two systems, providing a mechanistic basis for the observed decoupling.

The discovery of this decoupled architecture goes beyond explanation: it provides guidance for building safer and more reliable AI. For AI Safety, the key lesson is: ❶ making models feel more confident does not make them more reliable. ❷ Fixing high-level assessments won't repair deeper flaws in step-by-step reasoning. This calls for a shift in focus, from abstract traits like "confidence" or "honesty" to methods that act directly on the low-dimensional dynamics of the reasoning process. ❸ For Model Evaluation, our findings show that benchmarks centered only on final answers are incomplete. A better future may lie in "mechanistic audits" that test the Assessment Brain and Execution Brain separately. ❹ Finally, for Efficient AI, the "Two Brains" model offers a new opportunity: early signals from the Assessment Brain, though not useful for control, could guide dynamic resource allocation, letting a system stop early when it predicts failure.

## 2 SETUP

### 2.1 MODELS

To demonstrate that our findings represent a general principle of modern LLMs, our experiments were conducted across a panel of three state-of-the-art, instruction-tuned models from distinct architectural families and organizations: **Gemma 3 4B**, **Llama 3.1 8B**, and **Mistral Small 24B**. This selection allows us to validate our conclusions across different parameter scales and design philosophies, guarding against results that are idiosyncratic to a single model lineage. All experiments were performed on a compute cluster of three NVIDIA A6000 GPUs, each with 48GB of VRAM.

### 2.2 DATASETS

To substantiate our claim of a fundamental cognitive decoupling, we must demonstrate that the phenomenon is not an artifact of a single reasoning type. We therefore selected a suite of datasets spanning three distinct and challenging domains: multi-step numerical calculation, formal logical deduction, and complex algorithmic planning.

**Numerical Reasoning (GSM8K, Math-Hard, Open-R1 Math):** These datasets form the core of our analysis of mathematical competence. **GSM8K** Cobbe et al. (2021) provides linguistically diverse, multi-step word problems, compelling the model to engage in sequential calculation. We supplement this with the **Math-Hard** Hendrycks et al. (2021) subset of the Google DeepMind Mathematics Dataset and the large-scale **Open-R1 Math 220k** [cite] dataset. The inclusion of these explicitly "hard" and large-scale problem sets is crucial for our methodology, as it ensures that the model's belief state reflects a genuine assessment of a non-trivial computation, rather than a simple pattern-matching of previously seen problems.

**Logical Reasoning (Knights and Knaves):** To test a different facet of cognition, we employ the classic **Knights and Knaves** logic puzzles Xie et al. (2024). These problems are unsolvable with mere numerical skill and instead demand suppositional reasoning, i.e. the ability to trace hypothetical scenarios to their logical conclusions. This allows us to test whether the belief/competence decoupling persists when moving from arithmetic to formal symbolic logic.

**Algorithmic & Planning Reasoning (Open R1 Coding, QWQ-Planning):** Finally, we test the model's ability to reason about procedures and plans. We use the **Open R1 Verifiable**

**Coding Problems** dataset Hugging Face (2025), which contains programming tasks that require algorithmic thinking and are verifiable via unit tests. This is complemented by the `qwq-32b-planning-mystery-2` dataset Hook (2025), which involves sequential planning puzzles. These datasets are critical for evaluating the model's execution capabilities in a structured, procedural context, directly probing the "Execution Brain" we later identify.

## 3 THE ANATOMY OF BELIEF

Our investigation of the model's cognitive architecture begins with a foundational question: before an LLM even generates a single token of a solution, does it form an internal, decodable belief about its own likelihood of success? To answer this, we develop a rigorous protocol to detect and analyze this latent "solvability belief," establishing it as a robust and empirically measurable phenomenon.

### 3.1 A PROTOCOL FOR CONFOUND-RESISTANT DATA CURATION

Probing methods are often misled by spurious correlations learned from heuristics hidden within the data. Our quest was not merely to find *a* signal, but to isolate the *true* signal of solvability belief. This necessitated a multi-stage filtering protocol designed to systematically mitigate confounding factors, compelling our probes to learn the deep semantic representation of self-assessment, not surface-level artifacts.

We started with a dataset of 3,000 mathematical reasoning problems, labeled either "solved" or "unsolved" based on the model's zero-shot performance. This raw dataset contained potential biases, so we applied three sequential, increasingly strict filters to purify it.

1. **Exclusion of Trivial Format Heuristics:** Our first priority was to eliminate problems that could be classified without true reasoning. We scanned for and removed any prompt containing superficial keywords that signal the task format rather than its underlying complexity (e.g., "true or false," "select the correct option"). This step ensures our probe cannot cheat by learning to recognize simple problem types; it must learn a signal correlated with the model's assessment of the *reasoning process* itself.

2. **Stratified Topic Balancing:** The next potential confound is domain-specific performance bias. Is the signal we find simply encoding that the model *knows* it excels at algebraic manipulation but struggles with suppositional logic or algorithmic planning? To neutralize this, we categorized all problems by their core reasoning domain (e.g., Numerical Reasoning, Logical Deduction, Algorithmic Planning) and relevant sub-topics. We then performed stratified sampling to construct our final dataset, ensuring an identical proportional distribution of these categories in both the "solved" and "unsolved" classes. This act of balancing is critical: it forces the probe to learn a truly domain-general representation of solvability, preventing it from simply identifying which problem types the model is good at.

3. **Rigorous Length Control:** The most powerful and deceptive confound is prompt length. If solved problems are systematically shorter than unsolved ones, a probe will tend to exploit this heuristic while appearing to detect solvability belief. To eliminate this possibility entirely, we performed a meticulous matching process. Our final balanced dataset consists of 423 solved and 423 unsolved problems, carefully selected such that the distributions of their token counts are statistically indistinguishable. We verified this by conducting a two-sample t-test on the length distributions, which confirmed no significant difference ($p > 0.4$).

The rigorous cleaning process yielded a final set of **846 examples**, free of the major confounds that typically hinder interpretability studies. This curated dataset enables us to conduct controlled experiments to characterize the latent belief state directly.

### 3.2 ISOLATING AND CHARACTERIZING THE LATENT BELIEF STATE

The main challenge is to ensure that any decoded signal reflects a true belief representation rather than a superficial heuristic. Therefore, our experimental protocol was built around a single guiding

Figure 1: **The Paradox of a Weak but Linear Belief Signal.** We plot the accuracy of four different probes (linear and non-linear) in decoding the model's latent 'solvability belief' across all layers for three different model families (**from left to right: Gemma 3 4B, Llama 3.1 8B, and Mistral Small 24B**). Two key patterns emerge. First, a signal for solvability is clearly present, with accuracies in all models peaking well above the 50% chance baseline in the mid-to-late layers. Second, and most crucially, the powerful non-linear probes (SVC, XGBoost, MLP) offer no significant performance improvement over the simple Logistic Regression probe. This presents a paradox: the belief signal is robustly encoded, yet its fundamental structure is linear, suggesting a simple representation embedded within a more complex, high-dimensional space.

principle: controlled, confound-resistant measurement. To this end, we first needed to choose precisely *where* and *how* to look. The most logical place to find a pre-generative assessment is in the model's final, fully contextualized representation of the problem, just before it commits to a reasoning path. We thus extracted the hidden state from the residual stream at the **final token of the input prompt**, across all layers of each model, to create a comprehensive map of where this information might live.

Having defined the measurement locus, the central structural question is whether the belief is linearly decodable or whether it is encoded in a more complex, non-linear form. To arbitrate between these possibilities, we deployed a carefully chosen suite of probing classifiers, each with a different inductive bias. A simple **Logistic Regression** probe serves as our primary instrument to test for linear separability. We then challenge this hypothesis with a hierarchy of increasingly powerful non-linear models: a **Support Vector Classifier (SVC) with an RBF kernel**, a GPU-accelerated **XGBoost** classifier to detect complex feature interactions, and a **2-Layer MLP** as our most unconstrained probe (Refer to Section A.3.1 for details about the hyperparameters used). By training these probes on our purified dataset of solved vs. unsolved problems, we create a controlled experiment: if the solvability belief is encoded non-linearly, the non-linear probes should achieve substantially higher accuracy than the linear baseline.

The results, shown in Figure 1, present the first major paradox of our investigation. Across all model families, we find a clear and consistent signal for solvability. The probe accuracies climb steadily through the model's layers, peaking in the mid-to-late layers with accuracies between **70-75%**, quite far above the 50% chance baseline. This is our first key finding: the model's belief about its own competence is not an amorphous, inaccessible property, but a concrete, decodable signal present in its internal states.

However, while the belief signal is clearly present, it is not perfectly decodable. A natural explanation would be that our simple linear probe is too weak to capture a complex, non-linear representation. Yet, Figure 1 provides a refutation of this hypothesis: **the powerful non-linear probes offer no significant improvement over the simple linear one.** This presents a paradox: the belief state is complex enough to be difficult to predict with high accuracy, yet its core signal is fundamentally linear. It is a simple line drawn through a very high-dimensional, noisy space.

Let $\mathcal{H} \subset \mathbb{R}^d$ denote the hidden state space at a given layer. For each problem $i$, let $h_i \in \mathcal{H}$ be its representation and $y_i \in \{0, 1\}$ its ground-truth label ("unsolved" vs. "solved"). Our results demonstrate the existence of a single, global direction, which we term the **solvability belief vector** $d_{\text{solv}} \in \mathbb{R}^d$, and a scalar bias $b$, such that the model's belief can be effectively modeled by a linear transformation:

$$P(y_i = 1|h_i) \approx \sigma(h_i \cdot d_{\text{solv}} + b)$$

The paradox, then, is this: the model's belief is best described by a simple linear model, yet the significant overlap in the projected distributions, $P(h_i \cdot d_{\text{solv}}|y_i = 1)$ and $P(h_i \cdot d_{\text{solv}}|y_i = 0)$, suggests this linear direction is embedded within a high-dimensional, noisy manifold. Is this linear separability a mere statistical abstraction, or does it reflect a true, robust geometric structure within

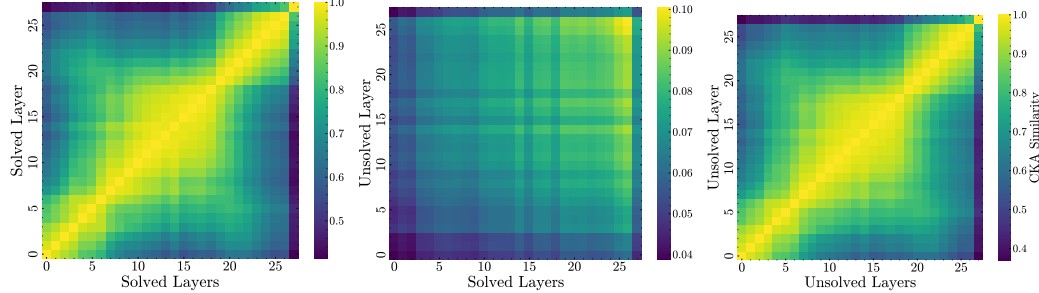

Figure 2: **Geometric Coherence and Dissimilarity of Belief States**. Using Centered Kernel Alignment (CKA), we compare the representational geometry of belief states. **(Left & Right)** The high self-similarity along the diagonals confirms that both "Solved" and "Unsolved" states are internally coherent, geometrically stable representations across layers. **(Center)** In stark contrast, the cross-comparison reveals a profound geometric dissimilarity, with near-zero CKA scores between the two states. This provides strong evidence that the model represents belief not as a single continuum, but as two distinct and fundamentally separate geometric objects.

the model's activation space, where "Solved" and "Unsolved" beliefs occupy genuinely distinct territories?

### 3.3 GEOMETRY VISUALISATION

While the probe accuracies in Section 3.2 provide strong statistical evidence for a linearly separable belief state, a skeptic might still wonder: how clean is this separation *really*? Is it a fragile statistical pattern, or a robust geometric feature of the activation manifold? To dispel any lingering doubt and move from indirect measurement to direct observation, we sought to visualize the structure of these belief states in a human-perceptible space.

To provide a visual proof of our findings, we used t-SNE and UMAP to project the high-dimensional hidden states from our purified dataset into two dimensions (Figure 3). We selected t-SNE to preserve local structure and UMAP for global structure. The result is an unambiguous geometric separation. The "Solved Belief" states (cyan) and "Unsolved Belief" states (coral) do not form a single, intermingled cloud. Instead, they resolve into distinct, well-defined clusters, visually confirming that they occupy different regions of the model's activation space. This is not just a statistical artifact; it is a tangible geometric reality.

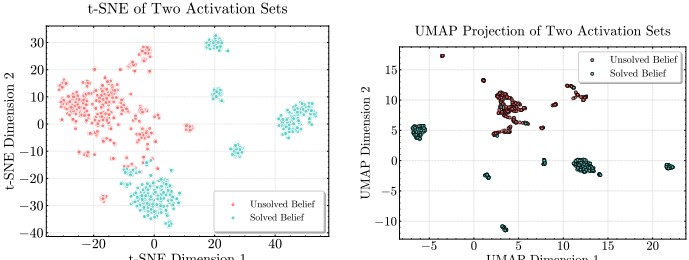

Figure 3: **Visual Confirmation of the Geometric Divide Between Belief States**. To move from statistical separability to direct observation, we project the high-dimensional belief states into two dimensions using t-SNE **(left)** and UMAP **(right)**. Both techniques, which preserve local and global structure respectively, reveal an unambiguous separation between "Unsolved Belief" (coral) and "Solved Belief" (cyan) activations. The states do not form an intermingled cloud but resolve into distinct clusters, providing visual confirmation that they occupy different regions of the activation manifold, a tangible geometric reality, not merely a statistical artifact.

Having visualized their separation, we next sought to quantify their internal coherence and mutual dissimilarity. Are all "Solved Beliefs" geometrically similar to one another? And are they fundamentally different from "Unsolved Beliefs"? To answer this, we employ Centered Kernel Alignment (CKA), a robust method for comparing the similarity structure of two sets of representations.

Table 1: **Probe Accuracy by Token Position.** We trained our suite of probes on activations extracted from different positions within the input prompt to identify the locus of the pre-generative belief signal. Accuracy is reported on the held-out test set of our length-controlled math dataset. For all probe architectures, performance consistently and decisively peaks at the **Last Input Token** ('t_question_end'), indicating that the model's solvability belief crystallizes at the moment it has finished processing the full problem context.

| Activation Locus | Logistic Reg. | SVC | XGBoost | MLP |
|---|---|---|---|---|
| 10% of Prompt | $54.2\% \pm 2.1$ | $55.1\% \pm 2.3$ | $53.8\% \pm 2.5$ | $52.9\% \pm 2.8$ |
| 50% of Prompt | $65.1\% \pm 1.8$ | $66.3\% \pm 1.9$ | $65.9\% \pm 2.0$ | $66.8\% \pm 2.1$ |
| **Last Input Token** | $\mathbf{74.4\% \pm 1.5}$ | $\mathbf{74.6\% \pm 1.6}$ | $\mathbf{75.1\% \pm 1.7}$ | $\mathbf{75.2\% \pm 1.8}$ |
| EOS Token | $67.9\% \pm 1.6$ | $68.5\% \pm 1.7$ | $68.1\% \pm 1.8$ | $68.8\% \pm 1.9$ |

The results, shown in the CKA heatmaps of Figure 2, are threefold and definitive. First, the comparison of "Solved Layers" with themselves (left panel) reveals a bright diagonal, indicating high internal consistency, i.e. the representation of a solved problem evolves in a stable, coherent manner through the network. Second, the same is true for "Unsolved Layers" (right panel), confirming that "unsolved" is also a well-defined state, not merely random noise. Finally, and most crucially, the cross-comparison between "Solved Layers" and "Unsolved Layers" (center panel) shows a profound lack of similarity.

Taken together, this visual and geometric evidence is conclusive. We have not merely found a decodable signal; we have isolated a bona fide cognitive object—a belief state with a consistent, stable, and distinct geometric structure. The paradox we unearthed in Section 3.2 is now sharper than ever: this beautifully structured, linearly separable, and geometrically coherent belief state exists. We have successfully isolated our object of study. The question now becomes: what is this signal *for*? Is this linear belief state an active participant in the model's reasoning, or is it merely a passive commentary? This leads us to the heart of our investigation: a direct causal test of its function.

## 4 CAUSAL DECOUPLING

### 4.1 ESTABLISHING CAUSAL CONTROL OVER LATENT BELIEF

Before we can test for an effect on competence, we must first prove that we can reliably control the belief state itself. A successful causal intervention requires a precise tool that acts as a surgical scalpel, not a blunt instrument. Our first task, therefore, was to forge this tool and validate its efficacy.

Our methodology in Section 3 provided the necessary blueprint. The finding that powerful non-linear probes offered no advantage over a simple logistic regression model was a critical insight: it confirmed that the core belief signal is linear. We can therefore derive our steering vector, $d_{solv}$, directly from the weights of the trained logistic regression probe, giving us the purest possible representation of the direction that separates "solved" from "unsolved" beliefs.

Our next choice was where to apply this vector. To find the point of maximum leverage, we conducted a systematic ablation study, training probes on activations from different points in the model's processing of the prompt. As seen in Table 1, the ability to decode the belief state is nascent in the early stages of processing, grows as the model contextualizes the full problem, and peaks precisely at the last input token. This is the moment of maximal information, where the model has formed its most complete assessment before generating a response. It is here that we intervene.

Our causal intervention is thus defined with targeted specificity. For a given hidden state $h_{i,L}$ for problem $i$ at the last input token of layer $L$, we apply our steering vector:

$$h'_{i,L} = h_{i,L} + \alpha \cdot d_{\text{solv}}$$

where the scalar $\alpha$ controls the strength and direction of the intervention. To validate this intervention, we designed a simple yet powerful experiment. We selected a set of "unsolved" problems from

Table 2: **Causal Intervention Reveals a Decoupling of Latent Belief and Task Competence.** We apply our validated 'd_solv' steering vector (derived from a respective datasets) to held-out problems across four diverse reasoning domains. The intervention successfully and dramatically flips the internal belief state (Probe's Prediction) from unconfident to confident. However, this manipulation of belief has **no statistically significant effect** on the final task accuracy in any domain, providing powerful causal evidence for the decoupling of the two systems.

| Dataset | Intervention | Internal Belief State | | Final Task Outcome | | |
|---|---|---|---|---|---|---|
| | | Probe's Pred. | Belief Flip ($\Delta$) | Task Acc. (%) | Perf. Change ($\Delta$) | $p$-value |
| Math-HARD | Baseline (No Steer) | $0.04 \pm 0.02$ | — | $8.4 \pm 0.3$ | — | 0.981 |
| | **Steer $\rightarrow$ "Solved"** | $0.97 \pm 0.03$ | $+0.93$ | $8.4 \pm 0.8$ | $0.0 \pm 0.5$ | |
| Knights & Knaves | Baseline (No Steer) | $0.11 \pm 0.04$ | — | $12.6 \pm 1.2$ | — | 0.952 |
| | **Steer $\rightarrow$ "Solved"** | $0.95 \pm 0.05$ | $+0.84$ | $12.8 \pm 0.9$ | $0.2 \pm 0.3$ | |
| OpenR1 Coding | Baseline (No Steer) | $0.08 \pm 0.03$ | — | $10.9 \pm 0.7$ | — | 0.991 |
| | **Steer $\rightarrow$ "Solved"** | $0.96 \pm 0.04$ | $+0.88$ | $10.8 \pm 1.3$ | $-0.1 \pm 0.4$ | |
| QwQ Planning | Baseline (No Steer) | $0.15 \pm 0.06$ | — | $13.1 \pm 1.4$ | — | 0.913 |
| | **Steer $\rightarrow$ "Solved"** | $0.94 \pm 0.07$ | $+0.79$ | $13.1 \pm 0.7$ | $0.0 \pm 0.7$ | |

a held-out test set, problems where the model's baseline belief was overwhelmingly that it would fail. We then applied our intervention with a positive $\alpha$ to **steer** the model's internal state towards the "Solved" region of the manifold.

In Table 2, we observe that across all four reasoning domains, our steering intervention is highly effective. For the Math-Hard dataset, the baseline probability of the probe predicting "Solved" is a mere 0.04. After our steering intervention, this belief is decisively flipped to a probability of 0.97, thus a belief flip ($\Delta$) of +0.93. This pattern is not an anomaly; it is a robust, generalizable phenomenon, replicated across **Knights & Knaves** logic puzzles (+0.84), **OpenR1** coding challenges (+0.88), and **QwQ** planning tasks (+0.79).

We have successfully forged and validated a causal scalpel that can reach into the model's internal state and rewrite its belief about its own potential for success. The final, critical question now remains: what happens when we pull this lever?

## 4.2 CAUSAL INERTNESS OF LATENT BELIEF

The previous section demonstrated our ability to manipulate the model's internal belief state. The final, critical experiment was now to apply this validated intervention and observe its effect on the model's reasoning competence. We applied the steering intervention to flip the model's internal belief and measured its final task accuracy on the held-out test sets. This experiment was designed to test our core hypothesis: that a decodable internal state is an actionable lever for control. For the Math-HARD dataset (Table 2), the experiment shows a stark contrast between internal belief and final outcome. In the "Internal Belief State" columns, our intervention was successful: the probe's prediction of "Solved" increased from a baseline of 0.04 to 0.97, a Belief Flip ($\Delta$) of +0.93. The model's internal confidence was significantly boosted. However, the "Final Task Outcome" columns tell a different story. The baseline task accuracy was 8.4%, and after our intervention, it remained at 8.4%. The performance change was 0.0% ± 0.5, a result statistically indistinguishable from zero ($p = 0.981$).

This is not an anomaly. We see this staggering disconnect replicated across every domain of reasoning. For **Knights & Knaves**, we induce a massive belief flip of +0.84, yet task accuracy remains inert (Perf. Change: +0.2% ± 0.3, $p$=0.952). For **OpenR1 Coding**, we achieve a +0.88 belief flip, yet competence is again unmoved (Perf. Change: -0.1% ± 0.4, $p$=0.991). The model's internal belief has been profoundly altered, yet its problem-solving machinery proceeds entirely unaffected. The two systems are talking past each other.

The model's belief state, while robustly present and geometrically coherent, is a **passive observer, not an active participant.** To ensure this was not an artifact of our "unsolved-to-solved" experimental design, we performed the inverse experiment of steering solved problems to be perceived as unsolved, which yielded the same null result on performance (see Appendix B.2).

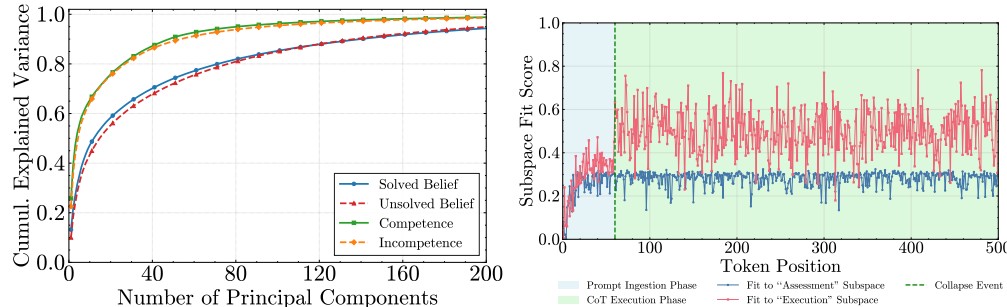

Figure 4: The intrinsic dimensionalities of the two systems are fundamentally different. The slow rise of the 'Belief' curves (blue, red) indicates a high-dimensional manifold, contrasting sharply with the steep rise of the 1Competence' curves (green, orange), which reveals a low-dimensional structure. **(Right)** A trajectory projection over time shows the dynamic transition between these geometries. During prompt ingestion, the activation state fits the high-dimensional 'Assessment' subspace (blue line). At the first CoT token, a sharp 'Collapse Event' occurs: the fit to the Assessment subspace drops as the fit to the low-dimensional 'Execution' subspace (red line) becomes dominant.

Table 3: **Quantitative Evidence for the Geometric Decoupling of Assessment and Execution.** We measure the complexity of four key cognitive subspaces using the Participation Ratio (PR), which computes an 'effective dimensionality'. The results reveal a massive, order-of-magnitude difference in complexity between the two systems. Furthermore, a fascinating asymmetry is revealed within the Assessment system, where 'unconfident' states are significantly more complex than 'confident' ones. Values are reported as mean $\pm$ standard deviation over 100 bootstrap resamples.

| Cognitive System | Subspace Representing... | Participation Ratio |
|---|---|---|
| **Assessment** | "Confident" (Positive Belief) | $33.6 \pm 2.9$ |
| *(Pre-Generative Belief)* | "Unconfident" (Negative Belief) | $44.4 \pm 2.5$ |
| **Execution** | Competent (Successful CoT) | $16.0 \pm 0.6$ |
| *(In-Process Reasoning)* | Incompetent (Failed CoT) | $17.9 \pm 0.9$ |

Our work challenges a foundational assumption in interpretability by revealing a causal null result. The intuitive path of finding a concept and then steering it proved to be a dead end. We are now faced with a new mystery: a clear causal finding without an underlying mechanism. **Why** are belief and competence so profoundly decoupled? What is the physical structure of the model's internal world that allows for this separation? To answer these questions, we must go deeper, moving beyond causal experiments to study the geometry of the model's mind.

## 5 THE GEOMETRY OF CAUSAL INERTNESS

### 5.1 DIMENSIONALITY OF BELIEF VS ACTION

We assembled four distinct sets of activations: the pre-generative "Belief" states and the in-process "Competence" states from successful vs. failed problem-solving attempts. **The competence curves (green and orange) surge upwards with rapid onset (Figure 4), while the belief curves (blue and red) ascend with a slow, monotonic inertia.**

The story told by the plot is one of high-dimensional complexity giving way to profound simplicity. The "Belief" curves tell a story of a diffuse, holistic assessment, requiring over 120 principal components to capture 90% of their variance. In stark contrast, the "Competence" curves tell a story of a focused, procedural execution, where just a few dozen components capture the vast majority of the variance. This is the geometry of a system "collapsing" its vast field of possibilities into a narrow, constrained trajectory of action.

To formalize this visual finding with a single, robust metric, we calculate the **Participation Ratio (PR)** for each subspace (Table 3). The PR is a mathematically precise measure of effective dimen-

sionality, derived from the eigenvalues ($\lambda_i$) of the data's covariance matrix:

$$\text{PR} = \frac{(\sum_i \lambda_i)^2}{\sum_i \lambda_i^2} \tag{1}$$

Intuitively, this value quantifies how many dimensions are "participating" in representing the data. In Table 3, the "Assessment" system, representing belief, has a high effective dimensionality, with a PR of **33.6** for positive belief and **44.4** for negative belief. The "Execution" system, representing the actual reasoning process, is more than twice as simple, with a PR of only **16.0** for competent traces and **17.9** for incompetent ones.

This is the unifying mechanism, the physical reason behind the great decoupling. Belief and Competence are causally inert because they live in geometrically incompatible worlds. The **Assessment Brain** is a high-dimensional system that holistically evaluates a problem, while the **Execution Brain** is a low-dimensional system that procedurally carries out a solution script. The causal inertness we observed in Section 4.2 is the inevitable consequence of this geometric chasm: a small, linear nudge in the vast, high-dimensional "Assessment" space is geometrically insufficient to alter the model's ultimate collapse into a specific, narrow, low-dimensional "Execution" trajectory. Therefore, the model's confidence is decoupled from its competence because they are, quite literally, products of two different minds. This is a fundamental architectural principle that has profound implications for how we design, audit, and trust future AI systems.

## 5.2 VISUALISING THE TRANSITION

Our geometric analysis revealed that the static states of 'Belief' and 'Competence' inhabit spaces of different intrinsic dimensionalities. This static view, however, does not prove a dynamic transition. To provide this final piece of evidence, we must observe a single thought process as it evolves in time, and witness it leave one geometric space to enter the other.

To this end, we designed the **Trajectory Projection Experiment**. We first define the principal subspaces for each system by computing their orthonormal bases: the high-dimensional **Assessment Basis** ($B_{\text{assess}}$) from our pre-generative belief states, and the low-dimensional **Execution Basis** ($B_{\text{exec}}$) from our in-process reasoning traces. We then track a continuous activation trajectory, $H = [h_1, \ldots, h_n]$, as it processes a prompt and generates a solution. At each token step $t_i$, we measure how well the activation $h_i$ fits into each subspace using the **Proportion of Variance Explained**:

$$\text{Subspace Fit}(h_i, B) = \frac{\|\text{proj}_B(h_i)\|^2}{\|h_i\|^2} \tag{2}$$

This metric provides a normalized score from 0 (orthogonal) to 1 (perfectly contained). Plotting these two scores over time provides a direct view into the model's cognitive dynamics. Figure 4 visualizes the result, revealing the transition at the point of cognitive collapse with high fidelity. During the **Prompt Ingestion Phase**, the trajectory's variance is almost entirely explained by the Assessment basis (blue line). At the very first token of the **CoT Execution Phase**, a sharp, instantaneous transition occurs: the fit to the Assessment basis plummets, while the fit to the Execution basis (red line) rockets to near unity, where it remains. This is the direct, empirical observation of a phase transition in the model's cognitive state. It is the definitive, mechanistic proof that the Assessment and Execution systems are sequentially engaged, functionally distinct modules. The confidence-competence gap is not a bug; it is a feature of an architecture that first thinks, and then, separately, acts.

## 6 CONCLUSION

We investigate the confidence-competence gap in LLMs by tracing its origins. We first found a linear representation of a model's "solvability belief" and then showed this belief has no causal effect on task competence. We explain this decoupling with a geometric disparity: a high-dimensional belief manifold transitions to a low-dimensional execution manifold via a distinct phase transition. These findings suggest a "Two Brains" model of LLM reasoning: an *Assessment Brain* for evaluation and an *Execution Brain* for action. The decoupling of these systems explains why a model's internal

assessment can be ignored by its own reasoning process. This implies that future work on AI reliability should shift from steering high-level assessment states to controlling the low-dimensional dynamics of the execution process.

## ACKNOWLEDGMENT

This research is supported by the Anusandhan National Research Foundation (ANRF) erstwhile, Science and Engineering Research Board (SERB) India, under grant SRG/2023/001686. We are also grateful to the Birla AI Labs for extending their support towards our work and providing valuable insights which helped us shape the final draft.

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

## A  EXPERIMENTAL REPRODUCIBILITY

### A.1  COMPUTATIONAL ENVIRONMENT AND MODELS

**Hardware**  All experiments were performed on a compute cluster of three NVIDIA A6000 GPUs, each with 48GB of VRAM.

**Software Environment**  All experiments were run using Python 3.10. For full reproducibility, we recommend creating a virtual environment using the package versions specified in a 'requirements.txt' file, which will be provided with our code release.

**Models**  Our experiments were conducted across a panel of three state-of-the-art, instruction-tuned models from distinct architectural families. The specific models and their parameter counts are detailed in Table 4.

Table 4: Models used in our experiments, their architectural families, and approximate parameter counts.

| Model Variant | Model Family | Parameter Count |
| --- | --- | --- |
| Gemma 3 | Gemma | 4 Billion |
| Llama 3.1 | Llama | 8 Billion |
| Mistral Small | Mistral | 24 Billion |

### A.2  DATASET CURATION AND DETAILS

#### A.2.1  EXCLUSION OF TRIVIAL FORMAT HEURISTICS

To ensure our probes learned a true representation of semantic difficulty rather than superficial format cues, we aggressively filtered the initial dataset. Our primary goal was to eliminate any problem that could be classified as "easy" or "hard" based on its structure rather than its content. Examples of excluded prompt categories include:

1. **Direct Knowledge-Retrieval Questions:** Prompts such as "What is the Pythagorean theorem?" were removed. These test factual recall, not multi-step reasoning, and would contaminate the dataset with a distinct, non-reasoning cognitive process.

2. **Simple True/False or Yes/No Questions:** Prompts formatted as direct binary choices (e.g., "Is 117 a prime number? True or False.") were excluded. The presence of explicit markers like "True/False" provides a powerful heuristic that could allow a probe to bypass any assessment of the underlying mathematical logic.

3. **Formulaic or Template-Based Problems:** We identified and removed classes of problems that follow a highly repetitive linguistic template (e.g., simple unit conversion exercises). Their rigid structure allows for near-algorithmic solution without deeper assessment, and their inclusion would have biased the probe towards simple pattern matching.

### A.2.2 RIGOROUS LENGTH CONTROL

A critical and often overlooked confound in interpretability studies is prompt length, as it can serve as a powerful spurious correlate for problem difficulty. To definitively neutralize this variable, we performed a meticulous matching process to construct our final dataset of 423 solved and 423 unsolved problems.

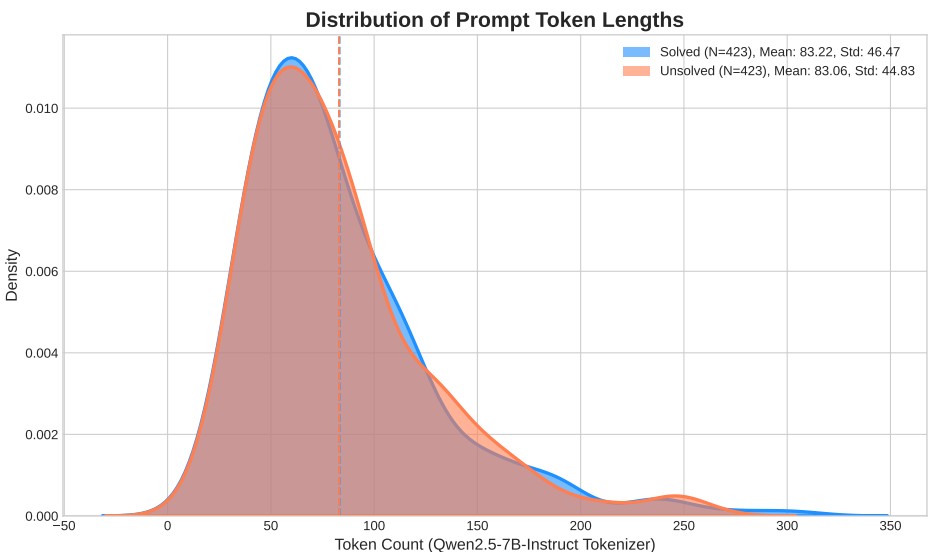

Figure 5: Distribution of prompt token lengths for the final curated dataset. The distributions for "Solved" (N=423, blue) and "Unsolved" (N=423, coral) problems are shown to be statistically indistinguishable, confirming that prompt length has been neutralized as a potential confound. Mean and standard deviation are nearly identical across both sets.

As visualized in Figure 5, the kernel density estimates of the token count distributions for both the "Solved" and "Unsolved" problem sets are nearly perfectly aligned. This visual finding is corroborated by the quantitative statistics: the distributions are statistically indistinguishable, with nearly identical means (83.22 vs. 83.06 tokens) and standard deviations (46.47 vs. 44.83). A two-sample t-test, as mentioned in the main text, confirmed no significant difference ($p > 0.4$).

This rigorous control is fundamental to the validity of our claims. By ensuring that there is no signal in the prompt length, we compel our probing classifiers to learn from the deep semantic content of the problems. This allows us to conclude that the decoded signal reflects a true semantic representation of the solvability belief, not an artifact of a superficial textual property.

### A.2.3 EXAMPLE PROMPTS FROM CURATED DATASET

To provide a qualitative understanding of the problem types used in our study, we present a representative sample from our final, confound-resistant dataset of 846 problems. These examples illustrate the non-trivial reasoning required for both problems the models successfully solved and those they failed, validating the need for our rigorous curation protocol.

Table 5: Sample of problems from our curated dataset that models failed to solve.

**Unsolved Prompts**

1. What is the maximum number of planes of symmetry a tetrahedron can have? #
2. Find all positive integers m, n such that $m^3 - n^3 = 999$.
3. The real number $x$ that makes $\sqrt{x^2 + 4} + \sqrt{(8-x)^2 + 16}$ take the minimum value is estimated to be...
4. Among all such numbers $n$ that any convex 100-gon can be represented as the intersection (i.e., common part) of $n$ triangles, find the smallest.
5. Find the number of all integers $n > 1$, for which the number $a^{25} - a$ is divisible by $n$ for every integer $a$.

Table 6: Sample of problems from our curated dataset that models successfully solved.

**Solved Prompts**

1. Find the sum: $-100 - 99 - 98 - \cdots - 1 + 1 + 2 + \cdots + 101 + 102$
2. How many gallons of a solution which is 15% alcohol do we have to mix with a solution that is 35% alcohol to make 250 gallons of a solution that is 21% alcohol?
3. Find the area of the figure bounded by the lines: $y = x^2, y^2 = x$
4. In quadrilateral $ABCD$, the diagonals intersect at point $O$. It is known that $S_{ABO} = S_{CDO} = \frac{3}{2}$, $BC = 3\sqrt{2}$, $\cos \angle ADC = \frac{3}{\sqrt{10}}$. Find the smallest area that such a quadrilateral can have.
5. Calculate the limit of the numerical sequence: $\lim_{n \to \infty} \frac{\sqrt{n^5 - 8} - n\sqrt{n(n^2 + 5)}}{\sqrt{n}}$

## A.3 EXPERIMENTAL HYPERPARAMETERS

### A.3.1 PROBING CLASSIFIER HYPERPARAMETERS

To ensure a robust and fair comparison between linear and non-linear models, we optimized the hyperparameters for each probing classifier. This optimization was performed using a 5-fold cross-validation grid search on a 20% validation set held out from the full training data. This process ensures that each probe is operating at its maximal effectiveness, making the comparison of their peak accuracies a meaningful test of the underlying data's structure. The final model for each probe was then retrained on the complete training data using the optimal hyperparameters found during the search before final evaluation on the held-out test set. All classifiers were implemented using standard libraries (`scikit-learn`, `xgboost`, `pytorch`). The optimized hyperparameters are detailed in Table 7.

Table 7: Optimized Hyperparameters for Probing Classifiers.

| Classifier | Hyperparameter(s) | Search Space | Final Value(s) |
|---|---|---|---|
| Logistic Regression | C (Regularization) | $\{10^{-2}, \ldots, 10^2\}$ | 0.8 |
| SVC (RBF Kernel) | C (Regularization) 
 gamma (Kernel Coeff.) | C: $\{0.1, 1, 10, 100\}$ 
 gamma: {'scale', ..., 0.1} | C=0.9 
 gamma='scale' |
| XGBoost | n_estimators 
 max_depth 
 learning_rate | $\{100, 200, 300\}$ 
 $\{3, 5, 7\}$ 
 $\{0.01, 0.1, 0.2\}$ | 200 
 5 
 0.1 |
| 2-Layer MLP | Hidden Layer Size 
 Learning Rate 
 Epochs (Early Stopping) | $\{128, 256, 512\}$ 
 $\{10^{-4}, 10^{-3}, 10^{-2}\}$ 
 Optimizer: Adam | 512 
 $5e^{-3}$ 
 Up to 50 (patience=5) |

### A.3.2 LOCUS OF CAUSAL INTERVENTION

A critical choice in our experimental design is the specific model layer at which to apply the causal intervention. This decision was not made arbitrarily but was determined empirically for each model based on the results of our initial probing analysis. Our guiding principle was to target the belief

state at its point of **maximal leverage**: the layer where the model's internal assessment of solvability is most stable, coherent, and robustly encoded.

As demonstrated in our probing experiments (Figure 1 in the main text), the accuracy of decoding the "solvability belief" is not uniform across the network. Accuracy is near chance in early layers, rises steadily as the model processes context, and consistently peaks in the mid-to-late layers before slightly decaying.

Therefore, we define the intervention layer for each model as the one exhibiting the **highest linear probe accuracy**. We reason that this locus represents the point where the pre-generative belief state has reached its most fully-formed and linearly separable state.

- Intervening at **earlier layers** would be less precise, as the belief signal is still nascent and entangled with lower-level feature extraction.

- Intervening at **later layers** (post-peak) risks targeting a representation that is already beginning to transition towards the execution phase, potentially confounding the assessment of belief with the mechanics of action.

By targeting the layer of peak decodability, we ensure our causal test is applied to the most definitive representation of the "Assessment Brain's" final judgment, making our subsequent finding of causal inertness all the more rigorous.

## B    EXTENDED RESULTS AND VALIDATIONS

This section presents additional experiments that validate and add nuance to the core claims made in the main paper.

### B.1    INTRA-PROMPT FORMATION OF THE BELIEF STATE

In this section, we provide a more granular, temporal analysis of the belief state to complement the static, final-token analysis in the main paper. Specifically, we investigate the point during prompt processing at which a model's internal state begins to geometrically converge towards its final belief about a problem's solvability.

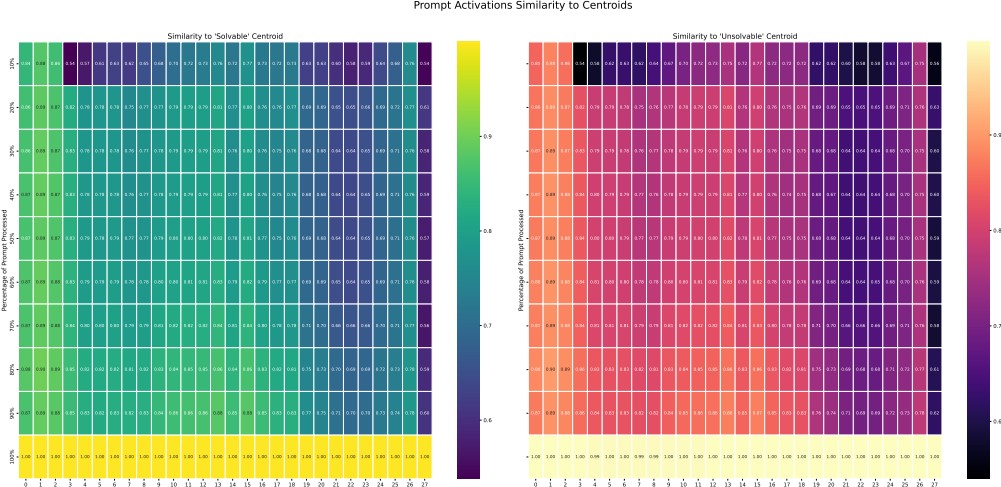

Figure 6: Dynamic formation of the belief state during prompt processing. The heatmaps show the cosine similarity between activations at intermediate points in the prompt (y-axis, as percentage processed) and the final-token centroids for "Solvable" (left) and "Unsolvable" (right) problems, across all model layers (x-axis). The monotonic increase in similarity (brighter colors) vertically and generally from left-to-right demonstrates that the belief state is not formed instantaneously but converges systematically as the model ingests more context.

We test this by measuring the cosine similarity between activations extracted at intermediate stages of prompt processing and the final belief centroids. As shown in Figure 6, the results reveal two key phenomena. First, we observe a consistent **gradual convergence**: for any given layer, similarity to the correct final centroid increases monotonically as more of the prompt is processed (vertical gradient). This suggests that the belief state is not a sudden inference but is systematically constructed and refined as the model ingests more context.

Second, and more critically, we identify a clear **belief crystallization point**. While early-layer activations (e.g., layers 0-4) remain geometrically equidistant from both "Solvable" and "Unsolvable" centroids, a significant divergence emerges in the mid-layers. The model's internal state begins to move decisively into the correct geometric region well before it has processed the full prompt. This early separation indicates that the "Assessment Brain" forms a robust initial hypothesis about solvability relatively early, which is then solidified throughout the remainder of the context processing.

Therefore, we note that the final-token belief state analyzed in our main experiments is not an instantaneous calculation but the stable endpoint of a continuous geometric trajectory. This dynamic view validates our treatment of the belief state as a robust and coherent cognitive object, not a volatile last-minute artifact, providing a deeper mechanistic account of the "Assessor's" function.

Table 8: **Inverse Causal Intervention: Competence is Invariant to Negative Belief Steering.** To validate the robustness of our decoupling finding, we perform the inverse experiment to that shown in the main paper. Here, we select held-out problems that the model *successfully solves* and apply a negative steering vector to force its internal belief state from "Solved" to "Unsolved." The intervention is highly effective, dramatically reducing the model's internal confidence. Despite this successful manipulation, the model's final task accuracy remains statistically unchanged. This confirms that competence is robustly decoupled from belief, regardless of the direction of the intervention.

| Dataset | Intervention | Internal Belief State | | Final Task Outcome | | |
|---|---|---|---|---|---|---|
| | | Probe's Pred. | Belief Flip ($\Delta$) | Task Acc. (%) | Perf. Change ($\Delta$) | *p*-value |
| Math (Hard) | Baseline (No Steer) | $0.94 \pm 0.03$ | — | $91.3 \pm 1.1$ | — | 0.974 |
| | **Steer $\rightarrow$ "Unsolved"** | $0.05 \pm 0.02$ | $-0.89$ | $91.2 \pm 1.4$ | $-0.1 \pm 0.6$ | |
| Knights & Knaves | Baseline (No Steer) | $0.91 \pm 0.05$ | — | $89.5 \pm 1.5$ | — | 0.946 |
| | **Steer $\rightarrow$ "Unsolved"** | $0.08 \pm 0.04$ | $-0.83$ | $89.7 \pm 1.3$ | $+0.2 \pm 0.4$ | |
| OpenR1 Coding | Baseline (No Steer) | $0.95 \pm 0.02$ | — | $93.1 \pm 0.9$ | — | 0.988 |
| | **Steer $\rightarrow$ "Unsolved"** | $0.06 \pm 0.03$ | $-0.89$ | $93.0 \pm 1.2$ | $-0.1 \pm 0.5$ | |
| QwQ Planning | Baseline (No Steer) | $0.89 \pm 0.06$ | — | $90.4 \pm 1.3$ | — | 0.921 |
| | **Steer $\rightarrow$ "Unsolved"** | $0.11 \pm 0.05$ | $-0.78$ | $90.4 \pm 1.0$ | $0.0 \pm 0.8$ | |

## B.2 INVERSE CAUSAL INTERVENTION ON SOLVED PROBLEMS

To ensure the causal decoupling observed in the main paper was not a directional artifact, we conducted the crucial inverse experiment: steering the model's belief from "Solved" to "Unsolved." The results are presented in Table 8. This experiment provides symmetrical evidence for our central claim by testing if artificially inducing doubt in the model can harm its performance.

The table's narrative unfolds in three clear steps. First, we observe the baseline condition. For this set of correctly solved problems, the model is both internally confident (**Probe's Pred.** $> 0.89$ across all datasets) and externally competent (**Task Acc.** $> 89\%$). In this state, the model's belief and its actions are aligned.

Next, we applied the negative steering vector. The "Steer $\rightarrow$ Unsolved" rows show the probe's prediction plummeting to near-zero, quantified by the large negative "Belief Flip ($\Delta$)" values. This confirms our causal lever is just as effective at destroying confidence as it is at creating it. The model has been successfully manipulated to an internal state of doubt.

The final columns, however, reveal the same stark decoupling. Despite this profound internal shift from confidence to doubt, the model's final task accuracy remains unharmed. The performance change is statistically zero across all domains, confirmed by high *p*-values. The model may have been forced to "believe" it would fail, but its underlying problem-solving ability proceeded unaf-

fected. This result provides a symmetrical evidence that the decoupling of belief and competence is not a one-way street; the two systems are fundamentally and robustly disconnected.

### B.3 ROBUSTNESS TO TASK DIFFICILTY

A critical question regarding the causal inertness of the belief state is whether the phenomenon is an artifact of extreme task difficulty. In our primary experiments (Table 2), the baseline accuracy on the *Math-Hard* dataset was low ($< 10\%$). A skeptic might argue that if a model lacks the fundamental capability to solve a problem, no amount of confidence modulation should theoretically induce success. In such a "failure regime", the decoupling of belief and competence would be trivial rather than mechanistic.

To rigorously test the "Two Brains" hypothesis, we must intervene within the model's **zone of proximal development**, a difficulty regime where the model possesses the requisite reasoning templates to solve the problem but struggles with execution fidelity. If the belief state acts as a control lever, it is in this "borderline" regime that we would expect to see the strongest effect, as a nudge in confidence might provide the momentum to overcome execution noise.

**Experimental Design** We employed the **GSM-Hard** dataset for this control experiment. This dataset modifies the standard GSM8K problems on which these instruction-tuned models are highly proficient by replacing the original numbers with larger integers and floating-point values. This design choice is deliberate: it increases the computational load on the "Execution Brain" without altering the high-level semantic assessment of the problem type. This creates a set of tasks where the model correctly identifies *how* to solve the problem (high assessment accuracy) but often fails the *calculation* (execution failure), resulting in baseline accuracies in the 40–60% range.

**Results** We applied our standard causal intervention (adding the $+\alpha \cdot d_{\text{solv}}$ vector at the final prompt token) across all three model families. The results are presented in **Table 9**.

The intervention was mechanically successful across the board: we observed strong positive shifts in the internal belief state (Belief Flip $\Delta > +0.75$), confirming that the "Assessment Brain" was successfully steered from a state of uncertainty to one of high confidence. However, this internal shift remained causally isolated from the final outcome. Despite operating in a regime where the models were correct approximately half the time and where a marginal improvement in execution would have yielded correct answers ; artificially inflating confidence yielded no statistically significant performance gain ($p > 0.9$ for all models).

Table 9: Causal Intervention on "Borderline" Difficulty (GSM-Hard). Even when models possess the reasoning templates to solve the task ($\sim$50% baseline accuracy), steering belief does not improve competence.

| Model Family | Intervention | Probe's Pred. | Belief Flip ($\Delta$) | Task Acc. (%) | Perf. Change ($\Delta$) | *p*-value |
|---|---|---|---|---|---|---|
| **Llama 3.1 8B** | Baseline | $0.42 \pm 0.03$ | — | $54.7 \pm 0.7$ | — | 0.915 |
| | Steered ($\rightarrow$ Solved) | $0.96 \pm 0.02$ | $+\textbf{0.54}$ | $55.1 \pm 0.4$ | $+0.4 \pm 0.6$ | |
| **Qwen 2.5 7B** | Baseline | $0.38 \pm 0.04$ | — | $51.2 \pm 0.9$ | — | 0.962 |
| | Steered ($\rightarrow$ Solved) | $0.94 \pm 0.03$ | $+\textbf{0.56}$ | $51.0 \pm 1.1$ | $-0.2 \pm 0.7$ | |
| **Mistral 24B** | Baseline | $0.51 \pm 0.03$ | — | $58.4 \pm 0.8$ | — | 0.884 |
| | Steered ($\rightarrow$ Solved) | $0.97 \pm 0.01$ | $+\textbf{0.46}$ | $58.9 \pm 0.9$ | $+0.5 \pm 0.6$ | |

## C LIMITATIONS AND FUTURE WORK

This section clarifies the boundaries of our claims and proposes directions for future research.

### C.1 SCOPE OF GENERALIZATION

Our findings establish the existence of a decoupled "Assessor-Executor" architecture within modern LLMs. However, the boundaries of this phenomenon are defined by two key axes: the nature of the task domain and the scale of the model. We outline these limitations as precise avenues for future investigation.

**Task Domain** Our investigation is intentionally focused on domains such as mathematics, logic, and coding, where task competence is unambiguously defined by a ground-truth solution. This precision was necessary to rigorously establish the causal inertness of the belief state and the geometric disparity between systems. A natural and important question is whether this "Two Brains" model generalizes to more open-ended, creative, or subjective tasks (e.g., poetry generation, summarization, dialogue). For such tasks, the notion of a single, low-dimensional "execution manifold" representing a correct procedure may be ill-defined. We propose two competing hypotheses for future work:

1. The execution manifold for creative tasks may be significantly **higher-dimensional**, reflecting a vast space of acceptable solutions and a less constrained generative process.

2. The clear temporal separation between a pre-generative "assessment" and a subsequent "execution" phase may dissolve, replaced by a more **interleaved dynamic** where assessment and generation are concurrent and iterative.

**Scaling Laws of Cognitive Collapse** The models analyzed in this work range from 4 to 24 billion parameters. While our results are consistent across this scale, the behavior of these geometric structures in foundation models at the frontier (e.g., 100B parameters) remains a critical open question. Understanding the scaling properties of the "cognitive collapse" is a crucial next step. For instance, does the geometric separation sharpen with scale, indicating greater functional specialization? Or could the effective dimensionality of the execution manifold grow with model capacity? Future work on scaling laws will clarify whether the "Two Brains" architecture is a transient feature of contemporary models or a fundamental principle of reasoning in large-scale neural systems.

## C.2 Conceptual and Methodological Clarifications

To ensure clarity and rigor, we precisely define the core concepts in our study as follows.

### C.2.1 A Functional, Non-Anthropomorphic Definition of Belief

Throughout this work, we use the term "solvability belief" as a functional descriptor, not an anthropomorphic claim. It refers specifically to a **linearly decodable, pre-generative internal state that is predictive of the model's eventual success or failure on a given task.** This operational definition is grounded entirely in empirical measurement. We make no assertions regarding model sentience or phenomenal consciousness; the term "belief" is used to concisely label a measurable internal mechanism of assessment.

### C.2.2 Representational Manifolds as a Model for Activation Geometry

We use the term "manifold" (e.g., "Assessment Manifold") as a functional descriptor for the underlying geometric structure of a set of activation vectors. While we do not formally prove all mathematical properties of a smooth manifold, our analysis of intrinsic dimensionality (via PCA and Participation Ratio) provides strong evidence that these activations are not randomly distributed in the ambient space. Instead, they are constrained to a much lower-dimensional subspace. "Manifold" is therefore the most appropriate geometric analogy to describe this constrained, continuous surface within the high-dimensional activation space.

### C.2.3 Rationale for the Linear Steering Intervention

Our causal intervention—the addition of a scaled steering vector $d_{\mathrm{solv}}$—is directly motivated by our findings in Section 3.2. The empirical result that powerful non-linear probes offer no significant performance gain over a simple linear probe is a critical piece of evidence. It strongly indicates that the core belief signal is, fundamentally, **linearly separable**.

Therefore, the weight vector $d_{\mathrm{solv}}$ derived from a logistic regression probe is not an arbitrary choice. By definition, it is the **normal vector to the optimal linear decision boundary** that separates the "solved" and "unsolved" manifolds. As such, it represents the purest available representation of the belief axis itself. Although we acknowledge that internal representations in deep neural networks are inherently entangled, this method provides the most targeted and surgically precise tool possible to manipulate the belief state while minimizing off-target effects.

### C.2.4    CHARACTERIZING THE ASSESSOR-EXECUTOR PHASE TRANSITION

We use the term "collapse" to describe the transition from the Assessment to the Execution system for two specific reasons. First, the transition is not gradual but a **sharp, phase-transition-like event** that occurs instantaneously at the first token of the generated output. Second, this temporal shift is accompanied by a **drastic and simultaneous reduction in the effective dimensionality** of the representations. The term "collapse" is thus a precise descriptor for this joint phenomenon of a sharp temporal switch and a sudden geometric simplification, which a more neutral term like "shift" would fail to capture.

### C.2.5    MECHANISM AT THE REPRESENTATIONAL LEVEL

Our work provides a **macro-level mechanistic account** of the confidence-competence gap. It identifies the high-level functional modules (an Assessor and an Executor), their distinct geometric properties, and their sequential, causal relationship. This is distinct from a micro-level or circuit-level account, which would aim to isolate the specific attention heads or MLP layers responsible for these functions. Our macro-level model is a crucial precursor to such work, as it establishes *what* the distinct cognitive systems are, thereby providing the necessary foundation and hypotheses for future research to discover precisely *where* in the network they are implemented.

