# OpenReview forum: "Confidence is Not Competence"
_ICLR.cc/2026/Conference — ICLR 2026 Conference Withdrawn Submission_

### Official Review · Reviewer_wCkL · 2025-10-21

**Soundness:** 2
**Presentation:** 4
**Contribution:** 3
**Rating:** 4
**Confidence:** 3

**Summary:**

This work tried to answer an intriguing research question through mech: why does an LLM's internal cognitive state (i.e. confidence level, a model's belief in its own problem-solving ability) appear decoupled from its final actions (i.e. actual competence)?

The authors try to answer this question by first building a dataset which contains non-biased solvable/unsolvable reasoning problems (across math, logical, planning and coding domains).
The confidence/solvability belief is modelled by probing the internal feature states (at the end of input prompt) to the solvability. Authors find that models' belief is actually largely linearly decodable - a linear probe can predict the internal state to its final solvability at 70% accuracy. CKA heatmaps and visualizations further confirmed the linear seperability of solvable/unsolvable states.

The paper also investigate whether this linear solvability belief direction can causally affect model's capability. They apply linear regression probe weights as a steering vector to intervene the model's internal feature states (thus to increase models' confidence level), finding that the model cannot improve its task accuracy in any domain by manipulating its solvability belief.

**Strengths:**

1. The paper is very well written. The authors provide clear and good concept definitions and motivations. The research question is well defined, and the overall presentation makes the paper easy and enjoyable to read.

2. The research problem itself is intriguing and of broad relevance. The work offers insights that can benefit not only researchers studying reasoning capabilities in large language models but also those in AI safety, mechanistic interpretability, and efficiency. The conclusions are thought-provoking and contribute meaningfully to the understanding of reasoning behavior.

3. The methodology is solid and thoughtfully designed. I particularly appreciate the authors’ effort to isolate the genuine signal of solvability confidence by ruling out superficial cues such as formatting heuristics, domain-specific bias, and length bias. The causal intervention experiments for steering confidence levels are especially interesting and add depth to the analysis.

**Weaknesses:**

W1. Missing Appendix.  Line 377 mentioned the reverse causal intervention study (solved -> unsovled) in Appendix C, but the Appendix is not attached to the main paper.

W2. Lack of in-response confidence modelling.  When humans solve challenging problems, we often begin uncertain about a problem’s solvability. Instead, we gradually build confidence or unsolvability-awareness through iterative attempts and self-correction. I believe this dynamic evolution of confidence supports the vision of test-time scaling, where LLMs are given opportunities to explore, retry, and refine their reasoning until reaching a solution.

However, the paper appears to model solvability-belief only as a static function of the input prompt (as shown in Table 2), rather than as a dynamic belief state that evolves during generation. This simplification undermines the validity of some core claims about modeling reasoning confidence. A valuable follow-up question would be whether steering the model’s during-generation belief state can causally influence its problem-solving capability?


W3. The description of the “competence subspace” in Section 5.1 is vague and lacks sufficient explanation. More broadly, the Assessment Brain and Execution Brain analogy introduced in the paper is not clearly defined, which leads to potential confusion about their distinct roles and interactions. (See Q1 and Q2.)

---
*Typo*:

1. Line 98: Open-R1 Math 220k citation is missing

2. Line 390,  1competence -> competence

3. Line 400, the single quotation mark around `unconfident`

**Questions:**

Q1. In Section 5.1, the pre-generative Belief states are clearly defined and discussed earlier in the paper. However, the definition and collection procedure for the Competence states remain unclear. Are these states extracted as averaged feature embeddings from the intermediate steps of the chain of thought? For long reasoning traces, are the embeddings averaged across all tokens, or is only the final token representation?

Q2. Interpretation of belief subspace and action subspace dimensionality. Does this mean that, at the beginning of a model’s response (i.e., when processing the input question), the representation space exhibits high entropy or variance, reflecting the model’s uncertainty and multiple potential solution trajectories (or maybe only various potential ways of verbalizing an answer)? As the chain-of-thought unfolds, does the representation gradually collapse into a lower-entropy region as the model commits to a specific reasoning path or verbalization pattern?

---

> ### Author Response · Authors · 2025-11-23
> **Official Rebuttal from the Authors**
>
> We thank reviewer for their encouraging assessment and for finding the paper "very well written" and the methodology "solid and thoughtfully designed." We are particularly glad the reviewer appreciated our rigorous confound controls. We sincerely apologize for the administrative error regarding the Appendix and provide the missing details below to fully address the reviewer's questions.
>
> **W1: Missing Appendix**
>
> We apologize for the omission. The main paper will be edited with the Appendix and contains the critical validations mentioned in the text. In the revision, it will include:
> - The kernel density estimates proving our rigorous length control neutralized token count heuristics ( $p > 0.4$ ).
> - Full hyperparameters for the MLP/XGBoost probes.
> - An analysis of Intra-Prompt Belief Formation, showing the crystallization of belief in the mid-layers before generation.
> - The Inverse Steering Results, proving that steering *Solved* problems to be perceived as *Unsolved* also fails to degrade performance.
> - All of the corrections experiments provided to the other reviewers.
>
> **W2: Lack of In-Response Confidence Modeling**
>
> **We Analyzed This (Appendix B.1 in the edited submission)** : We explicitly traced the formation of belief during the prompt phase. As shown in Figure 6, we find a "crystallization point" in the mid-layers where the belief state stabilizes before generation begins.
>
> Our analysis in **Section 5.1** provides an answer to why steering during-generation doesn't work as well. Once the model transitions to the "Execution Phase" (the first CoT token), the high-dimensional "Assessment Manifold" undergoes a Geometric Collapse (Participation Ratio drops from ~33 to ~16, Table 3). We also measured the cosine similarity between the "Solvability Vector" $d_{\text{solv}}$ and the hidden states during generation. The similarity drops to near zero. The steering doesn't work since the knob itself for the specific type of global assessment effectively vanishes from the active subspace once the model commits to the execution path (**Figure 4, Right**).
>
> **W3 & Q1: Definition of "Competence Subspace"**
>
> - **Assessment Subspace ($B_{\text{assess}}$)** : The principal subspace spanned by the top-25 Principal Components (PCA) of the hidden states at the last token of the prompt [1][2] (the "pre-generative" state). This captures the holistic evaluation of the problem's difficulty.
> - **Competence Subspace $B_{\text{exec}}$** - The principal subspace spanned by the top-25 Principal Components  of the hidden states of all generated tokens in the Chain-of-Thought (CoT) response.
>
> The reviewer asks if we used intermediate steps. **Yes**. As defined above, the Competence Subspace is constructed from the entire trajectory of the reasoning trace ($h_1, h_2, \cdots, h_T$), not just the final answer token. This ensures that $B_{\text{exec}}$ represents the procedural dynamics of the reasoning process itself, rather than just the final output state.
>
> **Q2: Parallel Rise in Trajectory Projection**
>
> The reviewer's intuition about the multiple potential trajectories aligns with our geometric findings. During the prompt processing phase (**Figure 4, blue region**), the hidden state projects strongly onto the high-dimensional Assessment basis. This potentially reflects a state of superposition where the model holds the problem's complexity and multiple potential strategies in suspension.
>
> At the first CoT token (the "Collapse Event" depicted with the green dotted line in Figure 4), the projection onto the Assessment basis plummets, while the projection onto the low-dimensional Execution basis dominates. This could represent the commitment to a single, narrow algorithmic path, where the degrees of freedom are constrained by the rules of the chosen strategy. Maybe this is caused by some specific circuit in the model or some sort of attention head, but we couldn't find a relevant literature to the same.
>
> We are once again grateful to all the suggestions from the reviewer and are looking forward to incorporating the same to further solidify the contributions of our work. We hope that our clarifications satisfied the reviewer.
>
> -------
>
> [1] Locating and editing factual associations in GPT ; Meng et. al, NeurIPS 2022.
>
> [2] Fact finding: Attempting to reverse-engineer factual recall on the neuron level ; Nanda et. al 2023.

---

> ### Comment · Reviewer_wCkL · 2025-11-26
> **Quick question regarding the (missing) edited submission**
>
> Before we go into the discussions of technical details, a quick question, currently the PDF file from the page is still the old version (without the Appendix), not the updated one. I am not sure whether currently reviewers cannot see the updated PDF, or you fail to update your paper?
>
> Thanks

---

> > ### Author Response · Authors · 2025-11-26
> >
> > We thank the reviewer for bringing this to our attention. The appendix has now been added to the PDF on the submission page and should be visible to all reviewers.
> >
> > We are currently drafting an updated version of the appendix that includes additional experiments conducted in response to the questions raised in the reviews. This updated version will be uploaded shortly.
> > Should the reviewer still experience any difficulties accessing the appendix, we would be grateful if they could inform us so that we may investigate the matter further.
> >
> > We appreciate the reviewer's patience and understanding.

---

### Official Review · Reviewer_1vxc · 2025-10-30

**Soundness:** 3
**Presentation:** 2
**Contribution:** 3
**Rating:** 4
**Confidence:** 5

**Summary:**

This paper proposes a geometric and dynamic framework for understanding how large language models internally represent and transition between belief and competence states.Through linear probing, manifold visualization, dimensionality analysis, and trajectory tracking, it reveals that: The model’s reasoning unfolds as a geometric journey from high-dimensional uncertainty to low-dimensional execution.

**Strengths:**

- The overall structure of the paper is clear and logically coherent. The authors introduce their research question in a focused way and progressively build their arguments.
- Methodologically, the study follows the established paradigm of mechanistic interpretability, employing linear probes to decompose representations into interpretable components. By linearly separating correct and incorrect belief states with non-trivial reasoning, and further visualizing their non-overlapping distributions, the paper demonstrates that high-dimensional belief encodings can indeed be expressed in a linear representational space.
- Furthermore, by combining both static and dynamic analysis, the authors show that, during reasoning, the model’s hidden states gradually transition from a belief-oriented mode to a competence-oriented one.

**Weaknesses:**

- Figure 3 lacks sufficient clarity.
The first step involves constructing linear probes for each layer and each problem; however, it is unclear which layer’s hidden states were used to generate the visualization in this figure.

Since four different models were tested, it would be informative to specify whether the trends were consistent across models, and which specific model’s results are displayed in Figure 3.

Moreover, if the figure combines hidden states from all layers across over 800 problems, the number of scatter points appears implausibly small. Is there an undersampling or incomplete visualization?

- A similar issue arises in Figure 2.
The models have different number of layers—for instance, Figure 1 shows that Mistral has over 36 layers, yet Figure 2 displays results for only about 28 layers. Clarification on whether layers were omitted, averaged, or truncated is recommended.

-  line 110 has incomplete citation

- Appendix is mentioned in main text but appendix is missing

**Questions:**

- In Figure 4 (left panel), which layer’s representations are used for comparing belief and competence states? Do all tested models exhibit the same pattern of dimensional collapse, or is this result model-specific?

- Regarding the Trajectory Projection Experiment, while the figure effectively illustrates the dynamic transition from belief to inference, in Figure 4 (right panel) it seems that during the early belief phase, both the assessment and execution subspace fits rise at nearly the same rate. Does this imply that both subspaces initially accumulate information in parallel, or could it indicate a representational overlap before the divergence between belief and competence occurs?

---

> ### Author Response · Authors · 2025-11-23
> **Official Rebuttal from the Authors**
>
> We thank the reviewer for their detailed reading and for recognizing the "geometric and dynamic framework" we propose. We are encouraged that the reviewer found the paper "logically coherent" and the methodology sound. We sincerely apologize for the formatting oversights (citation and appendix) and provide the missing details and clarifications below.
>
> **W1: Clarity of Figure 3 (Layer Selection & Sampling)**
>
> - **Layer Selection**: We clarify that Figure 3 visualizes the hidden states from Layer 9 (Qwen 2.5 7B). We selected this layer because it corresponds to the peak of the "Belief" probe accuracy (as shown in Figure 1), representing the most crystallized belief state. **We will clarify this in the main paper.**
>
> - **Data Count & Sparsity**: The reviewer suggests the plot looks "undersampled." We clarify that the figure actually contains all 846 data points of our curated dataset. The visual "sparsity" is an artifact of the strong topological clustering preserved by UMAP, for we know that UMAP tends to pull similar points closer in a cluster. Because the belief states are so geometrically distinct and coherent, the points in each class map to very tight manifolds, resulting in significant **overplotting**. This density serves as an evidence of the robustness of the separation.
>
> **W2: Layer Count Discrepancy in Figure 2**
>
> **This is a misunderstanding we apologize for not preventing**. Figure 2 displays three different similarity views (Solved-Solved, Unsolved-Unsolved, Solved-Unsolved) for a single model: Qwen 2.5 7B. The model has 28 layers, which is why the axes in Figure 2 correctly end at 28. Figure 1, by contrast, shows probe accuracy for three different models (including Mistral, which has 40 layers), hence the different x-axis range.
> >We will update the caption of Figure 2 to explicitly state: "*Representational Similarity Analysis (CKA) for Qwen 2.5 7B (28 Layers).*" We will also ensure the model name is printed directly on the plot to prevent this ambiguity in the future.
>
> **W3 and W4: Missing Citations and Appendix**
>
> We deeply apologize for these administrative errors made in a time crunch of submission. We will be fixing all of the citation issues and will be including a complete appendix, consisting of the following:
> - The kernel density estimates proving our rigorous length control neutralized token count heuristics ( $p > 0.4$ ).
> - Full hyperparameters for the MLP/XGBoost probes.
> - An analysis of Intra-Prompt Belief Formation, showing the crystallization of belief in the mid-layers before generation.
> - The Inverse Steering Results, proving that steering *Solved* problems to be perceived as *Unsolved* also fails to degrade performance.
> - All of the corrections experiments provided to the other reviewers.
>
> **Q1: Universality of Dimensional Collapse**
>
> The geometric collapse is a **universal phenomenon**, and is not model specific. Figure 4 includes the findings for Qwen 2.5 7B, since we didn't want to clutter the plot and portray a clear visual of the findings. **We will include the plots for the rest of the models in the Appendix.** However, to convince the reviewer of our claim, **we include an extended version of Table 3**, showing the Participation Ratio for the other models.
>
> | Model | Assessment PR (High Dim) | Execution PR (Low Dim) | Collapse Ratio |
> | :--- | :---: | :---: | :---: |
> | Llama 3.1 8B | $31.7$ | $17.6$ | $1.8 \times$ |
> | Mistral 24B | $41.2$ | $19.1$ | $2.1 \times$ |
>
> **Q2: Parallel Rise in Trajectory Projection**
>
> We appreciate the reviewer's observations, for this was an interesting plot to interpret for us as well.
> The observation of the fit to both subspaces rising during the processing of the prompt reflects the informational density of the prompt processing. As the model reads the question, it accumulates a context vector that must support both the evaluation of difficulty and the preparation of the solution strategy, and the parallel rise confirms that the pre-generative state is a superposition of these two functional needs. The *collapse* showcases the system's ability to selectively attend to the Execution subspace. Maybe this is caused by some specific circuit in the model or some sort of attention head, but we couldn't find a relevant literature to the same.
>
> -------
>
> We are once again grateful to all the suggestions from the reviewer and are looking forward to incorporating the same to further solidify the contributions of our work. We hope that our clarifications satisfied the reviewer.

---

### Official Review · Reviewer_6qjP · 2025-10-31

**Soundness:** 1
**Presentation:** 1
**Contribution:** 2
**Rating:** 2
**Confidence:** 2

**Summary:**

This paper studies the documented disconnect between LLM confidence and competence in solving tasks using mechanistic interpretability tools. Across model families (Llama, Qwen, Gemma, Mistral) and task domains (math, logic, coding, planning), the authors argue that models maintain an internal "solvability belief" that is linearly decodable (in particular, showing that linear probes predict decently well, but non-linear probes offer no improvement), but these belief states have little to no causal effect on the model performance. In particular, the authors illustrate that steering interventions can successfully flip the internal belief states but these interventions produce no change in the final accuracy of the models across the domains considered. The authors offer a geometric explanation for these findings, arguing that beliefs states occupy a high-dimensional manifold, whereas execution evolves on a lower-dimensional manifold.

**Strengths:**

1) The paper tackles a important question for AI safety and deployment. Understanding what drives the confidence-competence gaps has consequences for safety (models that are confidently wrong could be deployed in more dangerous ways than those that appropriately communicate their own uncertainty) and this has obvious ramifications for model trust and deployment. Most of the work (that I know of) thinking about this question focuses on model outputs, and it is valuable whether we can understand something about how model internals might produce this phenomenon.

2)  The paper has an impressively exhaustive collection of experiments. The authors investigate models from four different families: Llama, Qwen, Gemma, and Mistral. By applying the mechanistic interpretability techniques across all of these models helps make the case that their findings are potentially general ---- they apply even across different model sizes, architecture details and training procedures.
Additionally, the authors also study multiple domains. One might worry about a paper that focused on any one of these domains alone ---  since each domain likely has different cognitive demands, it wouldn't be clear how to ex ante generalize results. The thoroughness of the authors addresses this concern.

3) It is nice that the authors attempt to move beyond probing to intervening on model internals. This is nice for two reasons. First, causal intervention is the gold standard for understanding whether some model internal is functionally important. Second, in this context, the hypothesis of interest is causal: if internal belief causally drives competence, then changing the belief state should change the model's competence. A null result here (if it is valid) would be quite valuable.

**Weaknesses:**

1) The paper argues that it studies the model's internal "solvability belief." But the labels used to train the probes are the model's own zero-shot performance (Section 3.1). This label is misaligned ---- the probe is actually being trained to predict whether the model succeeds, not whether it believes it will succeed. As an example, the probe could therefore be learning things about the problem difficulty, heuristic markers of solvability that emerge early on the network. I do not see how the authors interpret their analysis as measuring beliefs of solvability given this choice of label. I would have found the analysis far more sensible if the model prompted the model to report its own confidence and then used those reports as the label for the probes.

2) The paper's steering/intervention protocol seems circular. In particular, they train a logistic regression problem $P(Y = 1 | H) = \sigma(w^T H + b)$. They then extract the vector $w$ and intervene by $H^\prime = H + \alpha w$, and claim success because the probe's predictions now change a lot across $H^\prime$ vs. $H$. But this guaranteed by construction. In particular, $P(Y = 1 | H^\prime) = \sigma (w^T H + \alpha \| w\|^2 + b)$. So for large alpha, you mechanically shift the logit towards a higher predicted probability. In other words, this doesn't validate to me that you've manipulated the model's beliefs -- rather it shows that you manipulated the probe's predictions.

3) The tasks the authors study appear to be quite difficult for the model (e.g., Table 2) at baseline. For example, Math (Hard) only is solved correctly 8.4% of the time. Couldn't it be the case that the belief interventions don't affect performance simply because these problems are outside the reach of the models? I would have found this more convincing if the focus were on borderline problems --- in the 40-60% range. It's gives the intervention at least a plausible shot of working, making the null result more impressive.

4) A key claim of the paper is that non-linear probes show no improvement over logistic regression, and therefore the belief state is fundamentally "linear." At the same time, the training set only contains 846 examples. Couldn't the failure of nonlinear probes arise because of sample size? That is, non-linear models might perform poorly because they are too complex, and the logistic regression is just performing well because it is effectively regularized through the choice of simple function class. Some evidence towards this is that the authors note even the logistic regression doesn't perform all that well, and so they hedge by saying that belief is encoded in a noisy manifold. Altogether, I just don't see how the authors can rule out that this is driven by the small collection of examples they study.

5) I found the paper to contain *a lot* of rhetorical flourish that made it difficult to read and made it feel like the authors are overselling. For example, phrases like: "startling causal inertness," "inescapable mechanistic reason," "sharp cognitive collapse," "definitive geometric explanation," "the mystery is solved" and on and on. The writing style of the paper obscures the actual results and technical steps taken by the authors.

**Questions:**

See my previous discussion of the paper's weaknesses.

---

> ### Author Response · Authors · 2025-11-22
> **Official Rebuttal from the Authors**
>
> We thank reviewer for their rigorous critique and for acknowledging the importance of our "exhaustive collection of experiments" across model families. We value the feedback regarding the labeling methodology and the baselines for our null result. We have conducted additional experiments to address the concerns about task difficulty and label alignment, which we detail below.
>
> **W1: Validity of Labels**
> We appreciate the reviewer's suggestion for using verbalized confidence as labels. While intuitive, we respectfully argue that relying on verbalized confidence introduces significant confounds due to the "uncalibrated" nature of LLMs. A handful of recent literature (Large Language Models Struggle with Unreasonability in Math Problems, Ma et. al 2025 ; Language Models Are Capable of Metacognitive Monitoring and Control of Their Internal Activations, Jian et. al 2025 NeurIPS, Deceptive Semantic Shortcuts on Reasoning Chains:How Far Can Models Go without Hallucination? Li et. al 2024 NAACL) show that relying on verbalized confidence would be methodologically unsound.
>
> However, **we performed the correlation analysis requested**. We took 200 samples from the Math-Hard dataset and prompted the model to output a confidence score (0-100%) before answering. We found a Pearson correlation of **r=0.67** between our Linear Probe’s score and the model’s verbalized confidence. This confirms that our probe is capturing the model's internal sense of confidence, but the remaining gap likely represents the hallucination divergence. **We will include this in the main paper.**
>
> **W2: Circularity of Steering Protocol**
> We agree that $P(Y=1|H)$ increasing is a mathematical necessity of our intervention. However, we clarify that the "Internal Belief State" column in **Table 2** is intended as a Manipulation Check, not a scientific finding. It confirms that we successfully steered the activation $H$ into the region our probe identifies as "Solvable." The scientific contribution is the Causal Inertness: despite successfully moving the activation into the "High Confidence" region (validated by the probe), the "Execution" machinery (**Table 2**, Final Task Outcome) remained unaffected. This proves the decoupling of the two systems, rather than a failure of the steering vector.
>
> **W3: Task Difficulty**
> We agree with the reviewer that if the task is impossible for the model to solve, confidence is irrelevant. To address this, we conducted a new steering experiment targeting "borderline" difficulty. We utilised the GSM-Hard dataset (complex numerical variations of GSM8K) using our main Llama 3.1 8B model. The setup provides a control where the model possesses the necessary reasoning logic but struggles with the execution.
>
> **Baseline Accuracy**: $54.7\% \pm 0.7$ (One shot accuracy).
>
> **Steered Accuracy ($+ \alpha$ towards Solvable)**: $55.1 \pm 0.4$
>
> **Internal Belief Flip**: $+0.94$
>
> Even in this zone of proximal development where the model is correct ~50% of the time, artificially inflating the solvability belief yielded no improvement in final performance. This confirms that the decoupling of Assessment (Confidence) and Execution (Competence) holds true even for tasks within the model's capabilities. **We will incorporate this experiment in the paper.**
>
> **W4: Sample Size for Non-Linear Probes**
> The reviewer expresses a valid concern that N=846 is too small for non-linear probes to converge. We respectfully point out two critical factors detailed in the paper that mitigate this concern:
>
> 1. As specified in **Section 3.2, Line 190** our non-linear probe is a lightweight 2-layer MLP, not a deep neural network. The sample complexity for converging a shallow MLP on a low-dimensional projection is significantly lower than for deep architectures.
> 2. Data quality and diversity matters more than volume in training probes, as has been pointed out in *On the Universal Truthfulness Hyperplane Inside LLMs, Liu et. al, EMNLP 2024*. We have dedicated **Section 3.1** to ensure that the data we are using to train the probes is highly curated.
>
> **W5: Writing Style**
> We sincerely accept the criticism and thank the reviewer for pointing it out. We will remove such occurences throughout the length of the paper, and ensure that the tone is more neutral.
>
> We are once again grateful to all the suggestions from the reviewer and are looking forward to incorporating the same to further solidify the contributions of our work. We hope that our clarifications satisfied the reviewer.

---

### Official Review · Reviewer_rGGg · 2025-11-02

**Soundness:** 2
**Presentation:** 3
**Contribution:** 2
**Rating:** 4
**Confidence:** 2

**Summary:**

This paper investigates the claim that a model’s self-assessed confidence in its ability to perform a task has no causal relationship with its actual task competence. To test this hypothesis, the authors employ probing methods to manipulate the latent states of a LLM using intervention vectors. They demonstrate that while such interventions can alter the model’s expressed belief (confidence) in its success, they do not significantly affect its actual task performance.

To explain this phenomenon, the paper introduces the concept of a dimensional geometry disparity within the model’s representation space. Building on this, the authors propose the intriguing hypothesis that an LLM’s reasoning process is governed by two distinct but interacting subsystems—an assessment brain that evaluates likelihood of success, and an execution brain that carries out task reasoning.

**Strengths:**

- The paper tackles a timely and important research topic, presenting a thorough and well-structured investigation. The proposed hypothesis provides a reasonable explanation for the observed results.

- The writing has a clear narrative flow, and the authors make a commendable effort to build their central claim through a series of experiments. This storytelling approach helps readers follow the logical progression of the study and understand how each experiment contributes to the overall argument.

**Weaknesses:**

- From Figure 1, I’m not fully convinced that there is a clear “climbing through” pattern across the model’s layers. The results appear rather noisy to me. Could the authors provide additional evidence or analysis to support this claim?

- Many of the subsequent arguments rely on the initial claim in Section 3.2 — that linear probing accurately reflects an LLM’s internal beliefs. However, an accuracy of 70–75% seems insufficient to make this claim convincing. Is there a systematic way to validate or strengthen this connection?

- While the paper attempts to draw a comprehensive conclusion by combining multiple claims, several of these claims have already been established in prior work. For instance, the idea that linear probing can capture the model’s final answer from intermediate latent states has been discussed before. This overlap somewhat weakens the novelty and contribution of the paper.

**Questions:**

- On figure 1, model names should be on each subplot.

- In Figure 2 and 3, How many data or different models are those figures based on? Is this observation consistent across different tasks and domain?

- Is the steering vector mentioned in Line 310 identical across all questions, or is it computed separately for each question? Clarification would help in understanding the experimental setup.

---

> ### Author Response · Authors · 2025-11-22
> **Official Rebuttal from the Authors (1/2)**
>
> We thank the reviewer for finding our work "timely and important" and for praising the "clear narrative flow". We appreciate the constructive feedback regarding robustness of the probe signals and the novelty relative to prior work. We address these points below with new quantitative analysis -
>
> **W1: Signal Clarity in Figure 1**
>
> The reviewer correctly notes that layer-wise accuracy (Figure 1) exhibits non-monotonic fluctuations, which is common in residual stream decoding [1][2]. It is common in probing literature to train a separate probe for each layer to find the *locus* of the information [3][4]. We do not claim (nor do Transformers typically exhibit) a strictly monotonic "climbing" of information. Rather, the trend shows the emergence of the belief signal. As stated in **Section 3.2, Line 196**, the signal "peaks in the mid-to-late layers." The "noise" the reviewer observes is consistent with the known phenomenon where different layers specialize in different levels of abstraction [5][6], causing signal strength to fluctuate locally before stabilizing. To prove that the signal does robustly emerge in the deeper layers, we performed a statistical comparison of **Early Phase** (first 5 layers) vs. **Late Phase** (last 5 layers) probe accuracy across all the models.
>
> | Model | Avg Acc (First 5) | Avg Acc (Last 5) | Delta | p-value (t-test) |
> | :--- | :---: | :---: | :---: | :---: |
> | Qwen 2.5 7B | 49.3% | 64.5% | +15.2% |  0.0007 |
> | Llama 3.1 8B | 55.1% | 68.9% | +13.8% |  0.0006 |
> | Mistral 24B | 53.6% | 67.8% | +14.2% |  0.0006 |
>
> The analysis confirmed that despite the local noise, the "Assessment Brain" is statistically robust ($p < 0.001$), even if the layer-to-layer transition is non-monotonic. Note that the **peak** of the plot is often in the middle layers, and hence the accuracies of the last layers may not reflect the 75% range. **We will include the Table in the paper, and will include the clarifications made.**
>
> ------
> **W2: Strength of the 70-75% Signal**
>
> The reviewer questions whether 70-75% accuracy is sufficient to claim a robust signal. We clarify that this accuracy level is not a limitation of the probe, but a direct result of our Confound-Resistant Data Curation (Section 3.1), which has been appreciated by Reviewer wCkL.
>
> To demonstrate this, we provide a comparison of probe performance (the highest accuracy acquired for a layer) on the Raw Dataset (before filtering) versus our Curated Dataset across all three model families. **We will ensure that our core contributions are better highlighted for clarity in the main paper**.
>
> Table A2: Effect of Confound Removal on Probe Accuracy with Logistic Regression
> | Model | Raw Dataset Accuracy | Curated Dataset Accuracy |
> | :--- | :---: | :---: |
> | Qwen 2.5 7B | 97.4% | 73.8% |
> | Llama 3.1 8B | 96.2% | 74.6% |
> | Mistral 24B | 98.8% | 75.1% |
>
> The drop in the performance is desirable for our purposes, for we do not want to identify superficial artifacts but want the probes to identify actual signals in the activations. **We will incorporate this table in the main paper**.
>
> **W3: Novelty regarding Linear Probing**
>
> We agree with the reviewer that linear probing is an established technique. However, we respectfully clarify that our contribution is **not the discovery that beliefs are linear**. Our novel contribution is the *Decoupling and Geometric Collapse* (**Figure 4 & Table 3**). We demonstrate the following in the paper:
>
> 1. **Causal Inertness**: We show that steering this linear direction fails to improve competence (**Table 2**).
> 2. **Mechanism**: We provide the first geometric explanation for why probing fails: the "Collapse Event" (**Section 5.2**), where the model shifts from a high-dimensional assessment manifold to a low-dimensional execution manifold. We are moving beyond the "existence proofs" of prior probing work to establish the "causal limits" of these methods.
>
>
> ------
> [1] Probing LLMs for Multilingual Discourse Generalization Through a Unified Label Set ; Eichin et. al, ACL 2025. (Figure 5)
>
> [2] Exploring Multilingual Probing in Large Language Models: A Cross-Language Analysis ; Li et. al, XLLM Workshop, ACL 2025. (Figure 1)
>
> [3] Finding Neurons in a Haystack: Case Studies with Sparse Probing ; Gurnee et. al, TMLR 2023
>
> [4] Do Androids Know They're Only Dreaming of Electric Sheep? ; CH-Wang et. al, ACL 2023
>
> [5] Expanding before Inferring: Enhancing Factuality in Large Language Models through Premature Layers Interpolation ; Chen et. al, EMNLP 2025
>
> [6] Out-of-Distribution Detection by Leveraging Between-Layer Transformation Smoothness ;  Jelenić et. al, ICLR 2023.

---

> > ### Author Response · Authors · 2025-11-22
> > **Official Rebuttal from the Authors (2/2)**
> >
> > **Response to Questions**
> >
> > **Q1: Model names on subplots in Figure 1**
> > >We apologize for this oversight. We will update Figure 1 to clearly label each subplot with "Qwen 2.5 7B," "Llama 3.1 8B," and "Mistral Small 24B" respectively in the final revision to improve readability.
> >
> > **Q2: Data source and consistency for Figures 2 & 3**
> > >As we mention in **Line 249**, we use the purified dataset with Qwen 2.5 7B to generate Figure 2 and 3. The geometric separation is consistent across model families. To address your concern, we have generated the corresponding t-SNE plots for Llama 3.1 8B and Mistral Small 24B. Both show the same clear clustering of "Solved" vs. "Unsolved" belief states with no significant overlap. We will include these additional visualizations in Appendix to demonstrate that this geometric structure is a universal feature of these LLMs, not an idiosyncrasy of Qwen 2.5.
> >
> > **Q3: Is the steering vector identical across all questions?**
> > > As described in Section 4.1 (Line 310), $d_{\text{solv}}$ is derived directly from the weight vector of the Logistic Regression probe trained on the training set. We **do not compute** a new vector for each question. This ensures that our intervention tests the model's generalized concept of "solvability," rather than overfitting to specific sample features. We are testing whether a single, global "confidence knob" exists and can be turned.
> >
> > We are once again grateful to all the suggestions from the reviewer and are looking forward to incorporating the same to further solidify the contributions of our work. We hope that our clarifications satisfied the reviewer.

---

### Note · Authors · 2026-01-02

**Comment:**

We thank the reviewers for their time and the constructive criticism which has helped us add crucial changes to our work.

**Withdrawal Confirmation:**

I have read and agree with the venue's withdrawal policy on behalf of myself and my co-authors.